# Exosome-like Systems: From Therapies to Vaccination for Cancer Treatment and Prevention—Exploring the State of the Art

**DOI:** 10.3390/vaccines12050519

**Published:** 2024-05-09

**Authors:** Hamid Heydari Sheikhhossein, Francesca Iommelli, Natalia Di Pietro, Maria Cristina Curia, Adriano Piattelli, Rosanna Palumbo, Giovanni N. Roviello, Viviana De Rosa

**Affiliations:** 1Department of Medical, Oral and Biotechnological Sciences, University “G. d’Annunzio” of Chieti-Pescara, 66100 Chieti, Italy; hamid.heydari@unich.it (H.H.S.); natalia.dipietro@unich.it (N.D.P.); mariacristina.curia@unich.it (M.C.C.); 2Villa Serena Foundation for Research, 65013 Città Sant’Angelo, Italy; 3Institute of Biostructures and Bioimaging, National Research Council, 80145 Naples, Italy; rosanna.palumbo@cnr.it (R.P.); viviana.derosa@ibb.cnr.it (V.D.R.); 4Center for Advanced Studies and Technology (CAST), University “G. d’Annunzio” of Chieti-Pescara, 66100 Chieti, Italy; 5School of Dentistry, Saint Camillus International University of Health and Medical Sciences, 00131 Rome, Italy; apiattelli51@gmail.com; 6Facultad de Medicina, UCAM Universidad Católica San Antonio de Murcia, 30107 Murcia, Spain

**Keywords:** exosomes, cancer vaccines: immune system

## Abstract

Cancer remains one of the main causes of death in the world due to its increasing incidence and treatment difficulties. Although significant progress has been made in this field, innovative approaches are needed to reduce tumor incidence, progression, and spread. In particular, the development of cancer vaccines is currently ongoing as both a preventive and therapeutic strategy. This concept is not new, but few vaccines have been approved in oncology. Antigen-based vaccination emerges as a promising strategy, leveraging specific tumor antigens to activate the immune system response. However, challenges persist in finding suitable delivery systems and antigen preparation methods. Exosomes (EXs) are highly heterogeneous bilayer vesicles that carry several molecule types in the extracellular space. The peculiarity is that they may be released from different cells and may be able to induce direct or indirect stimulation of the immune system. In particular, EX-based vaccines may cause an anti-tumor immune attack or produce memory cells recognizing cancer antigens and inhibiting disease development. This review delves into EX composition, biogenesis, and immune-modulating properties, exploring their role as a tool for prevention and therapy in solid tumors. Finally, we describe future research directions to optimize vaccine efficacy and realize the full potential of EX-based cancer immunotherapy.

## 1. Introduction

Cancer remains one of the main causes of death in world due to its increasing prevalence and difficulties for treatments. According to the International Agency for Research on Cancer (IARC), the incidence of cancer is increasing every year, and it is expected to rise from 20 million to 32.6 million in the next two decades [1]. The development of novel anticancer strategies is paramount for exploring therapeutic avenues. Investigating oncogene inhibition, with single [2,3,4] or combined treatment [5,6,7,8], and developing molecules capable of modulating processes involving specific DNA structures, such as quadruplexes [9,10,11,12,13] implicated in cancer, holds significant importance in advancing cancer therapeutics. Despite that, research is focused on developing innovative strategies for cancer prevention and personalized therapy, as tumor biology varies from for each different type. One of the novel anti-cancer approaches is antigen-based vaccination, utilizing tumor-specific antigens (TSA) and tumor-associated antigens (TAA). These molecules may originate from different classes, ranging from proteins constitutively overexpressed in cancer cells to proteins activated only during some step of tumorigenesis and cancer progression. However, beyond the identification of suitable antigens for vaccine development, the main challenges in this field are to find a suitable delivery system for immune activation and to establish a good antigen preparation process and avoid transplant rejection [14,15,16]. Exosomes are a new class of extracellular vesicles carrying several types of molecules in the extracellular space and playing a relevant role in cell-to-cell communication. They may be isolated from cell culture and biological fluids and can be identified by several type of markers, such as transmembrane and surface membrane proteins, as well as soluble proteins inside the exosome lumen. Remarkably, recent advancements in exosome research have unveiled their potential not only as delivery vehicles but also as mediators of tumor microenvironment (TME) communication and regulators of immune responses. In the immunotherapy era, as innovative anticancer treatment, exosomes may be a tool to develop preventive and therapeutic vaccines by acting as an antigen-presenting system or as a delivery vesicle to target TSA and TAA and stimulate an antitumor immune attack [17]. The intricate interplay between exosomes and the immune system offers new avenues for enhancing immunotherapy efficacy. Furthermore, by leveraging exosomes’ ability as therapeutic cargo, researchers can develop targeted and personalized treatment for each cancer type, administering exosomes originated from the same patient (known as autologous approaches). Ongoing efforts to engineer and functionalize exosomes to optimize their biodistribution hold promise for overcoming existing challenges in cancer vaccine development. With further preclinical and clinical validation, exosome-based vaccines may emerge as a helpful tool in the fight against cancer, opening new perspectives in the cure of aggressive tumors that are difficult to treat. Overall, in this review, following a brief introduction to the biological properties of exosomes and their role in cancer prevention and treatment, we explore their function in the immune response modulation and as a novel class of anticancer vaccines. In more detail, we organized the present manuscript in three main sections, in which we describe the landscape of cancer vaccination, thus exploring the different types of vaccines and their mechanism of action, and then we focused on the biological role of exosomes in causing immunity activation. Moreover, we highlight the potential applications of exosome-based vaccines in solid tumors and explore the challenges and perspectives in this field, including the variability of exosome cargo and potential drawbacks in utilizing them. Finally, we also discuss on the future perspectives of exosome-based therapy, addressing the need for identifying specific antigens and optimizing vaccine efficacy by also considering the age-related immune decline and tumor-induced immunosuppression, as well as the implications linked to immune tolerance and the dynamic nature of exosome compositions. 

## 2. Vaccines in Cancer and Exosome-Based Immunotherapy

### 2.1. Current Status of Anticancer Vaccines for Clinical Use

Vaccination, or immunization, has long been used to support the body in combating diseases [18]. A weakened form of a living organism carrying intact antigens is administered to trigger immunity against diseases like smallpox, pertussis, diphtheria, and bacterial infections, as well as toxins like tetanus and botulism. Ultimately, live attenuated forms of invading organisms can be administered to bolster an individual’s immunity against diseases such as polio, yellow fever, measles, and other viral infections [19,20].

There are different methods to design vaccines against pathogens, which are based on the structures and composition of pathogens, as well as the response of the immune system to specific antigens [21]. Vaccines have been classified into three categories, including classical vaccines, subunit vaccines, and multiepitope-based vaccines [22]. In summary, classical vaccines are categorized into live-attenuated vaccines and non-living or inactivated pathogens [23]. Both vaccines undergo harsh preparation conditions, which may degrade epitopes, which are the smallest antigen components recognized by immune cell receptors, thereby reducing the immune response [20]. Contemporary vaccination strategies focus on developing safer and more effective vaccines. Subunit vaccines use purified antigens, such as proteins or polysaccharides, to activate the immune system, either dependent on or independent of T-cells. Despite challenges in antigen identification, subunit vaccines offer reduced side effects and dosages compared to live attenuated vaccines [24]. The basic concept of multi-epitope vaccines is similar to subunit vaccines; nevertheless, in multi-epitope vaccines, various epitopes recognizable by multiple T-cell subsets are combined and delivered through an effective drug delivery system to combat tumors and viral infections [25]. Virus-like particles (VLPs) and nanoparticles are the two most well-known vehicles for delivering multi-epitope vaccines. Identifying suitable antigens and their dominant epitopes, along with developing effective delivery systems and predictive bioinformatics tools, remain challenges in this vaccination approach [26]. Currently, the FDA has approved both therapeutic and preventive cancer vaccines. Table 1 provides details of some recent vaccines and their specifications. In particular, it outlines FDA-approved vaccines against cancer, categorized into preventive and therapeutic types. Preventive vaccines include HPV and HBV, targeting various cancers. Therapeutic options like Sipuleucel-T for prostate cancer and BCG for bladder cancer are also FDA-approved. Additionally, clinical trials are underway for vaccines targeting pancreatic and oropharyngeal cancers [27,28,29,30,31,32,33,34].

Recently, the definition of exosomes and their potential role in cancer progression and affecting the immunity response has sparked significant research interest. Researchers are exploring the use of these extracellular vesicles (EVs) as a delivery system, leveraging their specific cargo, and as potential candidates for the development of new cancer vaccines [35]. We examine their role in immune modulation and explore their potential for vaccination. Lastly, we also discuss challenges and prospects in exosome-based vaccination.

### 2.2. Exosome-Based Vaccines and Anti-Tumor Immune Response

#### 2.2.1. Exosomes: Composition and Biological Role

EVs constitute a heterogeneous group of biological, nano-sized vesicles characterized by a bilayer membrane composition similar to the cells from which they originate [36]. They are released by almost all types of cells and are broadly categorized into three subgroups, including apoptotic bodies (ApoEVs), microvesicles, and exosomes. These classifications are based on their size, compositions, and origins. ApoEVs, released from apoptotic or dying cells, are larger in size (ranging from 1 μm to 5 µm) and are enriched in phosphatidylserine, DNA, and histones. Microvesicles, released from the plasma membrane, have sizes ranging from 200 nm to 2 µm and contain P-selectin, integrin, and CD-40 [37]. Exosomes, the smallest subset of EVs, with sizes between 30 nm and 150 nm, are secreted by multivesicular bodies (MVB). The composition of exosomes may vary between cells based on their conditions [38]. However, some of the specific markers for distinct exosome components are categorized in Table 2.

Nevertheless, beyond the protein and lipid markers mentioned in Table 2, exosomes contain various types of RNA, such as messenger RNA (mRNA), microRNA (miRNA), and non-coding RNA (ncRNA) [50,51,52]. Due to the downstream regulation ability of miRNA and their protection from degradation and stability, this type of RNA attracts more research attention to exosomes [53]. They also contain a wide variety of growth factor receptors, which are related to their cellular origin [54,55]. 

Exosomes are released through several consecutive steps (Figure 1A). In summary, early endosomes are formed by the inward budding of the plasma membrane. Communication between the Golgi apparatus and the early endosomes generates late endosomes or MVBs containing intraluminal vesicles, which are also called exosomes. Upon fusion of late MVBs with the plasma membrane, exosomes are released into the extracellular matrix (ECM). In another possible way, if these MVBs fuse with lysosomes, they will be degraded [56,57]. 

Exosomes are pivotal players in cell-to-cell communication because they transfer different cellular products to recipient cells in several body districts [58,59]. Table 3 shows some examples of the most common exosomes isolated from different cell types [60,61,62]. 

While the molecular intricacies of their synthesis remain elusive, despite a model proposed decade ago [63], it is evident that exosomes are likely formed via inward budding, establishing a membrane-enclosed compartment with a lumen similar to the cytoplasm [64,65]. These versatile vesicles find applications in diverse fields, such as diagnostics and therapeutics, and as biological biomarkers. Their ability to shuttle bioactive molecules between cells has opened avenues for immunotherapy, targeted drug delivery, and disease monitoring. Moreover, their stability in body fluids makes them promising candidates for non-invasive diagnostics and prognostics in various diseases, including cancer, neurodegenerative disorders, and infectious diseases [66,67,68].

Initially, the general belief centered around the role of exosomes in waste disposal for cell purification, aiming to remove unused and toxic materials. In 1889, Stephan Paget proposed the theory of “seed and soil”. In this theory, the invasion of tumor cells is supported by the interaction between tumor cells (as seeds) and their tumor microenvironments (TME, as soil) [69]. Subsequent studies revealed that exosomes, as new extracellular vesicles, play an extensive role in cell–cell and cell–ECM communications. After being released into the ECM as seeds, exosomes exhibit different behaviors based on their composition and the cell of origin [70]. For example, they can act as signaling complexes to stimulate target cells or transmit receptors, functional proteins, miRNA, and mRNA between cells [53]. As a result of their composition and ability to cross biological barriers and enter biological fluids, exosomes prove to be adaptable vehicles for transporting proteins, lipids, and oligonucleotides [71]. Figure 1B shows a representative 3D model of a complex between a human nuclear exosome and MTR4 RNA. Such types of complexes may be relevant for RNA biogenesis, transcription regulation, and favoring genome stability.

#### 2.2.2. The Biological Mechanism of Exosome-Based Vaccines for Tumor Suppression

As described in Table 3, exosomes may be enriched in proteins with immunological function and represent a potentially efficient antigen presentation system. They may also work as vectors for several tumor antigens, cause anti-tumor immune attacks, or produce memory cells, thus inhibiting tumor cells from escaping.

In particular, the first research on the use of exosome-based vaccines for tumor suppression was conducted by Zitvogel et al. in 1998 [72]. They employed dendritic cells (DCs) along with MART-1/MelanomA peptide for T-cell co-stimulation to isolate dendritic cell exosomes (DEXs), which expressed MHC-II and I and T-cell co-stimulatory molecules. Following this research, exosome-based vaccination for cancer immunotherapy became a more prominent and actively researched topic. The most widely utilized exosomes for therapeutic vaccines against cancer are DEXs, as they express various proteins on their surfaces, including integrins, HSPs, cytokines, receptors for cytokines, ligands for T- and B-cells, as well as lectins. Additionally, they possess features such as long validity and ease of engineering [22]. 

Exosomes released by cancer cells have the potential to either suppress or activate immune cells, while those released by immune cells generally serve to activate them [73]. Tumor-derived exosomes (TEXs), as mentioned in Table 3, have been reported to carry a diverse range of TAAs, such as proteins, ligands, RNA, and lipids, facilitating their uptake by recipient cells. A study by Muller et al. demonstrated the role of TEXs as mediators in delivering their cargo to T-cells. The researchers revealed that TEXs could upregulate inhibitory genes in conventional CD4^+^ T-cells (Tconv) by decreasing the expression of CD69 on their surfaces, subsequently impairing their functions. In contrast, increased expression of CD39 and adenosine production in regulatory T-cells (T-reg cells) was observed due to the downregulation of genes that regulate adenosine production [74]. With this broad spectrum of TAAs, TEXs can stimulate a wide range of T-cell colonies, making them a promising source for cell-free vaccination to induce anti-tumor immune responses. However, one of the most effective ways to harness this potential is through the combination of TEXs with accessory cells like DCs or adjuvants, such as immunoreagents, to efficiently generate T-cell responses, both in vitro and in vivo [75]. 

### 2.3. Innate and Adaptative Immune Response Mechanisms

Immunity refers to the body’s ability to defend itself against invading organisms [76]. The body has two intrinsic defense mechanisms against pathogens (Figure 2). The first method, intrinsic in the body, is known as innate immunity. This includes the phagocytosis of bacteria and other pathogens by white blood cells (damage-associated molecular patterns (DAMPs) released from dying cells, which lead to recognition of them), the skin’s resistance to organism attacks, and the presence of specific chemical factors capable of killing microbes [77]. The second type of immunity, known as acquired or adaptive immunity, is recruited when an organism enters the body, and the body responds with specific reactions to protect itself from that particular foreign organism [78]. There are two primary types of adaptive immunity. One involves the production of antibodies by B-lymphocyte cells, referred to as humoral immunity. The second type of acquired immunity is generated by the activation of T-lymphocyte cells, known as cellular immunity. These activated T-cells digest and eliminate microbes and invading factors [76,78,79,80,81].

Because acquired immunity is established after the entry of microbes into the body, it is necessary for the body to first identify foreign agents. Every external agent entering the body has specific protein or polysaccharide molecules on its surface that differ from other substances present on the body’s cell surfaces. These proteins or polysaccharides on the surfaces of invading agents are called antigens. It is important to note that antigens have a specific region in their structure, called the epitope, to which antibodies bind [78,79,80].

Upon encountering a foreign antigen, tissue macrophages phagocytize it, presenting the digestion products to lymphocytes. These lymphocytes then mature and transform into plasma cells, precursors of plasma B-cells. Plasma B-cells undergo rapid proliferation and become active plasma cells capable of producing antibodies at an extraordinary rate, providing defense against the foreign agent for several days [76].

Some activated B-lymphocytes do not transform into plasma cells. Instead, under the influence of factors secreted by helper T-lymphocytes (Th), they undergo rapid proliferation and give rise to clones of the same type of lymphocytes. These primed lymphocytes remain hidden in lymphoid tissues and are called memory cells. If the body encounters the same antigen again, memory lymphocytes specific to that antigen quickly respond and eliminate the invading agent. This way, the body develops immunity against that specific antigen [82,83].

Antigens bind to MHC proteins on the surfaces of antigen-presenting cells (APCs) in lymphoid tissues and then attach to T-lymphocytes. APCs include tissue macrophages, B-lymphocytes, and DCs. Generally, two types of MHC proteins exist, including MHC-I proteins that present antigens to cytotoxic T-cells and MHC-II proteins that present antigens to helper T-cells. Helper T-cells are crucial regulators of the immune system and produce a series of intermediary proteins called lymphokines to carry out their regulatory functions. In contrast to killer T-cells, which attack foreign agents, invaders themselves can attack and create pores in the membranes of target cells with the help of membrane proteins called perforins [78,79,80]. 

Exosomes likely play a crucial role in immune regulation, including immune stimulation and immune suppression through the existence and transmission of certain antigenic factors found within or on the exosome membrane [84]. The nature of the immune response mediated by exosomes is likely to vary depending on the cellular origin of the exosomes. As an illustration, exosomes derived from tumor cells have the capability to hinder immune cell responses by suppressing the cytotoxicity of CD8^+^ T-cells and NK cells. This inhibitory effect is primarily facilitated through the downregulation of NKG2D and the presence of TGF-β on the surface of exosomes originating from the tumor [73]. In the next section, we discuss the general concept of exosomes in immune regulation. Additionally, we explore the role of exosomes in cancer immunotherapy, including their function as anti-immune cancer vaccines.

### 2.4. Exosomes’ Role in Cancer Immune Response

The immunogenic effect of exosomes could be attributed to the presence of certain general antigens inside and on the surface of exosomes. For instance, RNA in the laminal part of exosomes has been shown to act as DAMPs, activating and triggering the innate immune response, specifically in DCs and macrophages [73]. Ismail et al. have found that macrophage-derived microvesicles could transfer miRNA to various target cells, including monocytes, fibroblasts, macrophages, and endothelial and epithelial cells, thereby inducing cellular differentiation in these target cells [85]. Researchers isolated exosomes from mast cells that were exposed to H_2_O_2_ to simulate oxidative stress conditions. They evaluated the effects of these derived exosomes on recipient cells, which were then re-exposed to oxidative stress. The researchers found that these exosomes had the capacity to induce tolerance to oxidative stress in recipient cells, reducing cell death. This result was associated with the role of exosomal mRNA, which varied under various conditions; its activity could also be attenuated by exposure to UV light [86].

As we mentioned earlier, there is a great connection between exosome function and the cells they originated from. In this regard, exosomes released by immune cells could play a vital role in immune regulations and also communication between the innate and adaptive immune system [87]. Therefore, it attracts more researchers’ attention to employ these vesicles for cancer immunotherapy, diagnosis, and drug delivery therapy [88,89].

The immune response stimulated by exosomes depends on the various molecules carried by them. As mentioned earlier in the text, APCs, such as DCs, present MHC antigenic peptides to T-cells. Exosomes derived from various cells have been shown to contain MHC proteins, enabling their introduction to T-cells through both direct and cross-dressing mechanisms. In the direct method, MHC proteins on the surface of exosomes directly interact with T-cells, leading to their activation. In the cross-dressing method, after APCs receive exosomes, they begin to process these antigens and present them to CD8^+^ T-cells. In a study conducted by Harvey et al., the direct and indirect mechanisms of CD8^+^ T-cell activation by cell-derived nanovesicles (CDNVs) were evaluated [90]. Initially, they activated DCs using IFN-ɣ and introduced the SIINFEKL peptide to produce CDNVs expressing the MHC-I ligand. These CDNVs could activate CD8^+^ T-cells through direct interaction or indirect activation via DCs acting as mediators, presenting MHC proteins to T-cells. However, there is not much research on the ability of exosomes to directly stimulate CD4^+^ T-cells. Additionally, studies have shown that the level of expression of stimulatory factors crucial for T-cell activation on the exosome surface is lower than that on the DC surface. For example, when comparing direct activation methods, the ability of exosomes is only 5 to 10% of that of DCs [91].

B-lymphocyte-derived exosomes display abundant MHC class I and II molecules and have the ability to activate CD4^+^ T-cells. Muntasell et al. conducted research to evaluate the connection between T- and B-cell involvement in immune response generation. They revealed that incubating naïve CD4^+^ T-cells with exosomes derived from B cells causes differentiation into INF-ɣ-secreting cells. In fact, MHC-II complexes were shown to be necessary for cytokine secretion from CD4^+^ T-cells [92].

A noteworthy example illustrating the role of cargo in the function and behavior of exosomes comes from those derived from macrophages. Macrophages can be classified into two distinct types, including pro-inflammatory M1 and anti-inflammatory M2 macrophages. M1 macrophages are known for secreting molecules such as reactive oxygen species (ROS), stimulating nitric oxide synthesis, and releasing immunostimulatory cytokines, including TNF-α, IFN-γ, IL-1 β, and IL-12, to combat cancer cells. On the other hand, M2 macrophages secrete immunosuppressive cytokines, including TGF-β, IL-4, IL-10, and IL-13, and reduce mannose receptors [93]. Exosomes derived from M1 macrophages have been shown to be taken up by macrophages and DCs in lymph nodes, potentially triggering both the innate and adaptive immune systems [94]. Another study revealed that exosomes released by IFN-γ-induced M1 macrophages could activate the NF-kB pathway, leading to an increase in the level of caspase-3. This results in the establishment of a proinflammatory environment [95].

### 2.5. Exosomes as Intelligent Drug Carriers for Cancer Immunotherapy

The mechanisms involved in cancer immunotherapy have already been explained in the previous section (adaptive immunity). Various approaches exist for cancer immunotherapy, including blocking immune checkpoints, the regulation of TME, and cancer vaccines (see next section). Several types of drugs approved by the FDA for blocking immune checkpoints, such as inhibitors for CTLA-4, PD-1, and PD-L1, are currently in clinical use. However, their efficacy is moderated by the lack of targeted delivery [96].

Over the last two decades, with the advent of nanotechnology as a new field of study and the outstanding properties of nanoscale materials in medical applications, the use of nanomaterials as drug delivery systems has increased. These nanoscale materials have demonstrated several advantages, including enhancing the efficacy of anti-cancer drugs through targeted delivery and increasing their bioavailability [97]. Exosomes are capable of overcoming physical barriers and have lower immunogenicity compared to conventional nanoparticles, making them great candidates for a cancer drug delivery system [98]. In this context, a more detailed exploration of the potential application of exosomes in delivery and immunotherapy against tumor-draining lymph nodes (TDLNs) was presented in a study conducted by Panpan Ji et al. The researchers utilized a type of smart exosomes with engineered functionalities, including CD62L (L-selectin as a marker for TDLNs) and OX40L (a gene for activating T-cells and inhibiting regulatory T-cells) on its surface. The study revealed that the engineered exosomes were not only specifically delivered to TDLNs but also exhibited inhibitory activity against T-reg cells. This inhibition resulted in the suppression of tumor development, in contrast to control exosomes [99].

## 3. Exosome-Based Vaccines in Solid Tumors

A common approach to developing a cancer vaccine involves targeting the immune system to inhibit cancer cell growth. The vaccine is formulated by combining antigens that can differentiate cancer cells from normal cells with an adjuvant that enhances the immune response [100]. This adjuvant can serve as a delivery system for antigens, or even a drug that stimulates immune responses [101]. With the discovery of exosomes and the recognition of their remarkable characteristics, researchers have come to realize that exosomes can be utilized as both adjuvants and antigens simultaneously [102].

### 3.1. Lung Cancer

According to the World Health Organization (WHO) in 2022, more than 2,480,000 people were diagnosed with lung cancer, and approximately 1,817,000 died from the disease, making it the first most prevalent and harmful type of cancer in 2022. Like other types of cancer, immunotherapy plays an important role in the treatment of this cancer. However, only a subset of patients responds to the blocking of immune checkpoints, highlighting the need for new methods to diagnose early stages and emphasizing the significance of effective treatment for improved overall treatment quality [103,104]. Therefore, exosomes as a new class of vaccination have garnered significant attention as potential candidates for improving the efficacy of treatment for lung cancer patients. 

In 2012, Yaddanapudi et al. employed embryonic stem cells (ESCs) in conjunction with granulocyte-macrophage colony-stimulating factor (GM-CSF) as prophylactic vaccines for Lewis lung carcinoma (LLC). They observed that 70–80% of mice exhibited 90 days of tumor-free survival. The researchers found increased levels of memory CD8^+^ T-cells and Th1 cytokines in vaccinated mice, contributing to the prevention of tumor development upon re-challenge with live LLC cells 60 days later. This result indicated a long-term memory response. Furthermore, a decrease in the levels of Myeloid-derived suppressor cells (MDSCs) in lymphoid organs corresponded to the immunosuppressive response induced by tumor cells [105]. In agreement with the previous report, in an in vivo study conducted by Meng et al., LLC cells were injected into mice through the tail vein to induce metastasis. The study aimed to evaluate the effectiveness of embryonic stem cell-derived exosomes (exo-ESCs) expressing GM-CSF as an adjuvant for new prophylactic vaccines, targeting the inhibition of metastatic lung tumors. The researchers observed a reduction in immune suppressor cells, such as Treg and MDSCs, after ES-exo/GM-CSF vaccination. Simultaneously, CD8^+^ cytotoxic T-cells, which are associated with antitumor immunity, began releasing cytotoxic cytokines such as IFN-ɣ and TNF-α [106].

### 3.2. Breast Cancer

Breast cancer is a heterogeneous type of cancer classified based on histologic characteristics or receptor expression. Some types of breast cancer may be recognized by the overexpression of transmembrane proteins, such as HER2, HER3, EGFR, c-MET, and MUC1 [107]. Among all the vaccine-based methods, cell-based vaccines have several specific features that make them of great interest to breast cancer researchers. Exosomes are considered a type of vesicle that can promote tumor progression by transferring receptors to recipient cells and have also been studied for immune therapy in breast cancer [108]. In a study, researchers isolated exosomes from HER2-positive cells and observed that trastuzumab emtansine (T-DM1) specifically bound to the exosomes derived from HER2-positive cells rather than MCF-7 HER2-negative cells. These data suggest that exosomes could be a suitable delivery system for transferring drugs against receptors that are overexpressed on breast cancer cells [109]. Several studies have also investigated the role of exosomes as vaccines for HER2-positive tumors by targeting and stimulating CD4+ and CD8+ T-cells, as well as promoting long-term immunity through CTL memory cells [110,111]. In another study, researchers made genetically engineered breast cancer cell-derived exosomes that overexpress α-lactalbumin (α-LA) and loaded them with immunogenic cell death (ICD) inducers, ELANE (catalytically active neutrophil elastase) and Hiltonol (TLR3 agonist), simultaneously. In this vaccine, the ELANE showed selective anti-cancer properties with low toxicity toward not tumor cells. At the same time, TLR3 leads to the activation and maturation of DCs, which then enhance anti-tumor reactive CD8+ T-cells generation [112].

### 3.3. Pancreatic Cancer

Surgery is the only treatment option for pancreatic cancer as a result of late diagnosis and the early spread of metastasis [113]. In this regard, it is necessary to identify suitable biomarkers for early-stage diagnosis and develop strategies to overcome and target their potential for metastatic behavior.

A study evaluated the efficacy of vaccinating tumor exosome-loaded DCs in combination with gemcitabine, ATRA, and sunitinib, well-known chemotherapy drugs for pancreatic cancer. It was observed that these exosome-based vaccines could influence the activity of MDSCs and decrease their activity in tumor infiltration [114]. Other studies have shown that exosomes play a significant role in the communication between pancreatic cancer cells and immune cells [115]. Consequently, combining chemotherapy with drugs that inhibit the secretion of exosomes could be a great strategy to increase patient response to treatment and improve survival. Therefore, exosomes not only could be suitable for delivering immunomodulatory agents but also, by inhibiting their secretion from cancer cells, could trigger an anti-cancer immune response and prevent cancer cells from reprogramming TME [116]. Nevertheless, this approach could have disastrous consequences, as it is also possible to prevent normal cells from releasing exosomes. Consequently, a useful and safer method involves blocking a specific protein in exosomes that performs a significant function in tumor progression or promotes metastasis. A good example of this approach was in a study conducted by Hoshino et al., in which they found that by knocking out or inhibiting a specific factor that was overexpressed in pancreatic cancer cell-derived exosomes, the metastasis of pancreatic cancer cells to the lungs and liver could be inhibited [117].

In a clinical trial study, researchers used atezolizumab, an anti-PD-L1, and autogene cevumeran together with mRNA-lipoplex nanoparticles. After this novel treatment, they found that vaccine-expanded T-cells comprised up to 10% of all blood T-cells, re-expanded with a vaccine booster, and included long-lived polyfunctional neoantigen-specific effector CD8^+^ T-cells [33]. It is also promising and possible to target macrophages for the treatment of pancreatic cancer. Researchers have also shown that miRNA expressed from bioengineered exosomes derived from pancreatic cells could reprogram and alter the macrophage phenotype, inhibiting the immunosuppressive role of M2 macrophages [118]. 

### 3.4. Ovarian Cancer

The three most well-known applications of exosomes in ovarian cancer and other types of cancer are exosome-based drug delivery systems, the prevention of cell–cell communication by inhibiting exosome biogenesis, and exosome-based immunotherapy [119]. CA125 is a specific marker for the diagnosis and progression of ovarian cancer, although it exhibits low sensitivity and specificity. As exosomes have demonstrated a remarkable role in transferring markers from cancer cells to their TME, exploring exosomes as carriers of tumor-specific markers for cancer progression or treatment represents a promising research area [120]. Studies have shown that human adipose mesenchymal stem cells (haMSC) have a relationship with ovarian cancer cells, which could act as a suppressor or even an activator for cancer cells. In a study, exosomes derived from haMSC have demonstrated an inhibitory role on ovarian cancer cells by blocking cell cycles and inducing apoptosis signaling through increased levels of molecules, such as BAX, CASP9, and CASP3. These exosomes have also shown downregulation of the BCL_2_ protein. When RNase was added to the exosomes, it was found that they could not inhibit cancer cell progression and proliferation, indicating the role of miRNA as an inhibitory factor in the isolated exosomes [121]. 

Another research study has shown that exosomes derived from ascites in patients with ovarian cancer could be employed as novel immunotherapy. These exosomes express a variety of proteins, including HSP-70, HSP-90, MHC-I, HER-2/Neu, and Mart-1, which play a role in T-cell activation through DCs presenting antigens [122]. Despite the understanding of the immunotherapy against ovarian cancer, currently there is no approved method for the immune therapy of ovarian cancer [123].

## 4. The Challenges and Prospects of Exosome-Based Vaccines

The cargo of exosomes varies from cell to cell and depends on cell origins. Based on their cargo, they could have various roles in cell–cell communication. Take cancer-derived exosomes as an example, which could increase cancer progression by preparing secondary niches for cancer cells and also transfer cargo to immune cells for the immune escape of cancer cells. Because of this variable and dual behavior, numerous scientists have attempted to identify the target molecules of exosomes that are responsible for their specific roles in cancer progression. Therefore, one of the emerging research areas for cancer treatment could be blocking the expression of these specific proteins. However, these markers may also serve as dual-purpose adjuvants and antigens for cancer vaccines. The identification of these specific agents presents a considerable challenge. If they lack specificity for cancer cells and fail to discriminate effectively, potential side effects may arise. Another hot research area related to exosomes is their application for cancer immunotherapy to develop therapeutic exosome-based vaccines for cancer treatment.

However, despite all the mentioned potential applications of exosomes, there are some drawbacks when researchers want to use exosomes as cancer-based vaccines. As has been proven before, the ability of the immune system decreases with age. Because exosome-based vaccines rely on the immune system, this could be a significant problem. Another possible challenge is that studies have shown higher activities of immunosuppressor cells for larger tumors, which could counteract the immune response induced by exosome-based vaccines. It should also be considered that the immunogenic response generated by exosomes is less than that of normal vaccines. This immune tolerance could impact cancer cells, causing them to alter their exosome compositions and exacerbate the situation.

## 5. Conclusions

In conclusion, exosome-based vaccines represent a promising frontier in immunotherapy, offering a multifaceted approach to cancer prevention and treatment. These nanoscale vesicles possess unique attributes that make them ideal candidates for delivering antigens and modulating immune responses. By harnessing exosomes’ ability to carry tumor-specific antigens and interact with immune cells, researchers are exploring innovative strategies to stimulate anti-tumor immune responses and enhance therapeutic efficacy. Despite challenges such as immune tolerance and antigen specificity, ongoing research endeavors seek to overcome these obstacles and maximize the potential of exosome-based vaccines. Looking ahead, the applications of exosome-based cancer immunotherapy hold great promise. Continued investigation into exosome composition, biogenesis, and immune modulation will deepen our understanding of their therapeutic potential. Moreover, advances in exosome engineering and drug loading techniques will further enhance their utility as intelligent carriers for cancer immunotherapy. Engineered exosomes, having superior properties to natural ones in terms of stability, long-term circulation, and targeted function, may be used as a tool to improve the development of precision medicine in the field of cancer vaccines, thus preventing or inhibiting tumor growth and progression. Synthesis protocols for engineered exosomes, which involve the genetic engineering of parent cells to express desired surface proteins or loading methods for therapeutic cargo, such as electroporation or sonication, facilitate their customization for specific biomedical applications [124]. For example, exosomes engineered to display specific antigens on their surface can selectively modulate immune response. Additionally, engineered exosomes loaded with therapeutic payloads, such as chemotherapy drugs or RNA-based therapeutics, can enhance drug delivery efficiency and minimize off-target effects. In summary, with ongoing research and innovation, exosome-based cancer vaccines may soon become a cornerstone of comprehensive cancer care, opening new possibilities for cancer treatment.

## Figures and Tables

**Figure 1 vaccines-12-00519-f001:**
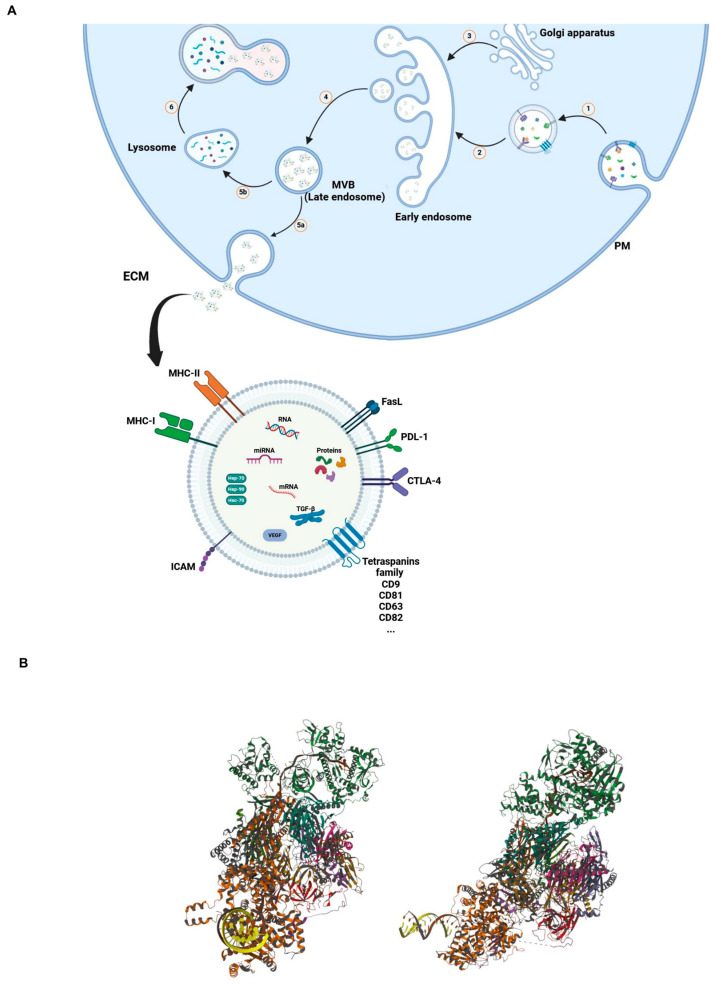
(**A**) Mechanism of synthesis and release of EVs. Inward budding of plasma membrane (PM) toward cytosol led to the formation of early endosome (1 and 2); communication between early endosome and Golgi apparatus formed and released late endosome or multivesicular bodies (MVB) (3 and 4); upon fusion of MVB with PM, exosomes are released into the ECM (5a); combination of MVB with lysosome result in degradation of MVB in cytosol (5b and 6). (**B**) Three-dimensional structural views of the human nuclear exosome-MTR4 RNA complex. (The structure is publicly available at the link: https://www.rcsb.org/3d-view/6D6Q/1, accessed on 6 March 2024).

**Figure 2 vaccines-12-00519-f002:**
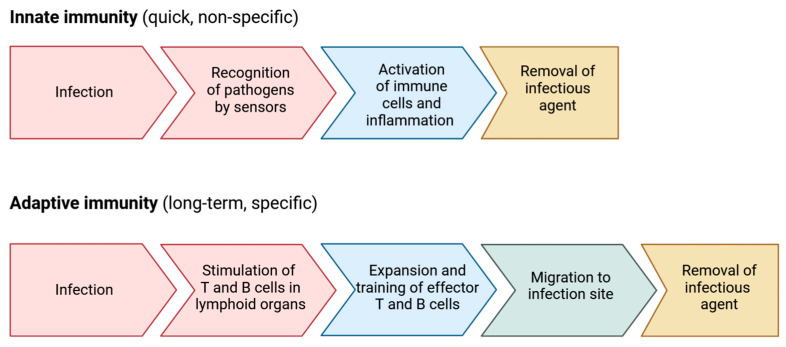
Immune defense mechanisms.

**Table 1 vaccines-12-00519-t001:** FDA-approved therapies and clinical trials for cancer vaccines.

Type of Vaccines	Availability	Name	Vaccine Strategy	Type of Antigens	Adjuvant	Type of Cancer	Refs.
Preventive	FDA approved	HPV (Gardasil Gardasil9 Cervarix)	Multiepitope VLP	Protein	yes	Cervical, Head and Neck, Anal, Penile, Vaginal, Vulvar	[27]
		HBV	Subunit VLP	Protein	yes	Liver	[28]
Therapeutic	FDA approved	Sipuleucel-T (Provenge)	Autologous	Protein Prostatic acid phosphatase (PAP)	Yes	Prostate	[29]
		BCG	Classical	PAMPs and DAMPs	No	Bladder	[30]
		Nadofaragene firadonevec (Adstiladrin)	DNA vector	BCG	No	Bladder	[31]
		T-VEC (Imlygic)	Classical	Live attenuated HSV-1	Yes	Melanoma	[32]
	Clinical Trials	Phase II clinical	Multiepitope Nanoparticle	mRNA and checkpoint inhibitors	Yes	Pancreatic	[33]
		Phase II clinical	Multiepitope	HPV (Protein based)	Yes	Oropharyngeal	[34]

**Table 2 vaccines-12-00519-t002:** Membrane and lumen exosome markers.

Content	Molecular Type	References
Membrane Markers	Tetraspanins family (CD9, CD81, CD63, CD82)	[39,40]
	Immunomodulatory (MHC-I, MHC-II, PDL-1)	[41,42,43,44]
	Lipid raft (PS, Sphingolipids, Cholesterol, Ceramide)	[45]
Luminal Space	Chaperones (HSP-70, HSP-90, HSC-70)	[46]
	MVB-Associated Proteins (ALIX, TSG101, Clathrin, Flotillin-1)	[47]
	Signaling proteins (HIF-α, TGF-β, cdc-42, VEGF, ARF-1)	[48,49]

**Table 3 vaccines-12-00519-t003:** Most common isolated exosomes, their molecular profiling, and applications.

Exosome Types	Molecular Profiling	Applications
Tumor-derived exosomes (TEXs)	miRNA, mRNA, DNA, lipids, and proteins promoting cancer invasion and progression	cancer diagnostics, prognostics, and therapy monitoring
Immune cell exosomes	MHC-I, MHC-II, CD1, CD47, adhesion molecules, costimulatory proteins (CD86), integrins	immunomodulation and autoimmune disease therapy
Neural cell exosomes	neurotrophic factors	neurodegenerative disorder diagnostics and therapy

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
