# Peer review of "Exosome-like Systems: From Therapies to Vaccination for Cancer Treatment and Prevention—Exploring the State of the Art"

_vaccines, 2024, doi:10.3390/vaccines12050519_

Round 1

Reviewer 1 Report

Comments and Suggestions for Authors

I have read the interesting review by Sheikhhossein and colleagues on “exosomes’ composition, biogenesis, and immune-modulating properties, exploring their role as a tool for prevention and therapy in solid tumors”. I recommend publication after some major issues have been addressed:

- The flow of the review should follow the definition cited above. The logic of the different sections and subsections is sometimes confusing (for example, please combine sections 2, 4.3, and 5)

- At the end of section 1, please introduce all the following sections of the review.

- Section 2: I believe it is not necessary to spend almost an entire chapter on general FDA-approved vaccines against cancer. Please shorten it by summarizing the details and references in Table 1. Moreover, consider combining this section with section 5.

- Section 3: it’s a crucial section for this review. Thus, it should be expanded by reporting examples of the most common exosomes (possibly in a Table). Please summarize the compositions (most common membrane’ o lumen’ antigens) as well as the applications that will be cited later in the following sections.

- Please better summarize Figure 2 in section 4, giving just the essential information of the two types of immune response necessary to understand the following two sections (immunotherapy and vaccine). Consider a separate section for this.

- The resolution of both Figures 1 and 2 (text and scheme details) is inadequate. Please reformat them.

- A citation of “engineered exosomes having superior properties to the natural ones” was included by the authors in the conclusion section. Please add examples and definitions of both exosome types, including the most important synthesis protocols and the known applications up to now.

Comments on the Quality of English Language

I recommend a revision of the English structure of some sentences

Reviewer 2 Report

Comments and Suggestions for Authors

My opinion is that the manuscript must be reconsidered taking into account important issues, listed below:

1. It seems that the aim of this review is indicated in the section 2 (vaccines and cancer): - rows 119-123. I think that it is important to specify the aim at the end of the Introduction.

2. Row 120 - "...exosomes as a mechanism for immune response..." - What does this statement mean?

3. Table 1 - If the title of the Table is FDA approved therapies, why did the Authors include therapies in Phase II CTs?

4. (!!!) I`ve detected a lot of self-citations (by De Rosa, Palumbo, Lomelli) in the manuscript, more than 10% of all references.

5. Minor English editing is required

6. The title of Section 4 is unclear for me (Mechanism of Immune system??).

7. Figure 1 A - too small resolution, can`t see exosome structure and markers within ECM.

8. Figure 2 - too small resolution, can`t see the details. 

9. T helpers - Please, use abbreviation throughout the text (Th).

10. 5.2 Section - different line and paragraph spacing

....

Comments on the Quality of English Language

-

Round 2

Reviewer 1 Report

Comments and Suggestions for Authors

I thank the authors for the revision of their article. Although some of my points were precisely answered, the review still presents an overall inaccurate structure. Here are my further comments:

- The newly added summary-sentence at the end of the introduction section perfectly summarizes all the topics of the review. Please, follow that order using a section for each topic;

- Section 2 “Vaccines in cancer and exosomes-based immunotherapy” has to be revised:

1) The subsection “2.2. Exosomes: definition, composition and biological role” does not belong here

2) The subsection “2.6. Exosomes based vaccines and anti-tumor immune response” should be moved to the vaccine section since it contains the biological mechanism of exosome-based vaccines for tumor suppression

Thus, I suggest to focus this section only on exosomes’ role in cancer immune response and immunotherapy;

- Sentence in Raws 122-125 “Exosomes present advantages like blood-brain barrier penetration, biocompatibility, stable encapsulation, and targeted delivery. We examine their role in immune modulation and explore their potential for vaccination” is completely out of context. Please move it. Moreover, the role of exosomes in immunomodulation is later reported only in Table 3 and never discussed in detail. Please, explain what this means;

- If you added “prevention and treatment” to section 3 (Exosomes-based vaccines in solid tumors) you should divide the reported exosome-based vaccines for both categories, as in Table 1.

Reviewer 2 Report

Comments and Suggestions for Authors

The Author`s responded to all comments.

Round 3

Reviewer 1 Report

Comments and Suggestions for Authors

All comments were answered, thanks.